# What role should EBU catheters play in the interventional approach to anomalous right coronary arteries? A case series

Mikias Legesse Gebremedhin[1], Milan Sigdel[2], Zhao Ruixue[1], BinBin Du[1], Yanzhou Zhang[1]*

1 Department of Cardiology, The First Affiliated Hospital of Zhengzhou University, Zhengzhou, China,
2 Department of Intervention Radiology, The First Affiliated Hospital of Zhengzhou University, Zhengzhou, China

* fcczhangyz@zzu.edu.cn, fccdubb@zzu.edu.cn

## Abstract

### Background

Percutaneous coronary intervention (PCI) for anomalous right coronary artery (ARCA) remains technically challenging due to variable ostial anatomy. The Extra Back-Up (EBU) catheter, although designed for left coronary interventions, may offer advantages in ARCA PCI, but its performance across anatomical subtypes is not well defined.

### Methods

This single-center retrospective case series included 17 patients (2019–2024) with ARCA from the left coronary sinus or with high take-off anatomy, in whom an EBU catheter was used. Fourteen patients underwent PCI, one had fractional flow reserve (FFR)-only assessment, and two underwent diagnostic angiography alone. Anatomical subtypes were: Type A (ostium above left coronary artery [LCA]), Type B (below LCA), Type C (midline), or high take-off (≥10 mm above sinotubular junction). Primary outcomes were procedural success (stable engagement, full device delivery, no catheter exchange) and safety.

### Results

Procedural success with the EBU catheter was 100% in Type A (2/2) and 83.3% in Type B (5/6) when used as the initial guide. In Type C and high take-off anatomies, success was achieved in one of two cases each. EBU was also employed as a second-choice catheter in five patients(one Type A, three Type B, one high take-off) after failed engagement with standard guides catheters. No catheter-related dissections occurred. One minor in-hospital complication (5.9%) and one patient required unplanned re-intervention within 30 days.

**Data availability statement:** All relevant data are within the paper and its Supporting information file.

**Funding:** The author(s) received no specific funding for this work.

**Competing interests:** The authors have declared that no competing interests exist.

## Conclusions

In this descriptive series, the EBU catheter was successfully employed as a primary or rescue tool in a majority of select ARCA cases, with the highest success in Types A and B. These observations, while not comparative, suggest it is a viable option in the operator's arsenal for these challenging anatomies and warrant further prospective evaluation.

## Introduction

Anomalous coronary arteries represent a spectrum of congenital variations that pose unique challenges for percutaneous coronary intervention (PCI) [1–3]. Anomalous coronary artery from the opposite sinus (ACAOS) is a rare congenital condition, affecting approximately 0.2–2.0% of the population [4,5]. Among these ARCA from the left sinus of Valsalva (LSOV) constitutes a particularly complex subgroup, with an estimated prevalence of 0.26–0.5% in angiographic studies [6,7]. The clinical significance of these anomalies stems from their association with myocardial ischemia, arrhythmias, and sudden cardiac death, particularly in young athletes during physical exertion [8,9].

Anatomically, ARCA from the LSOV can be classified into main subtypes based on ostial location relative to the left coronary artery (LCA), including Type A (ostium above LCA), Type B (below LCA), and Type C (midline origin), in addition to high take-off anomalies [10,11] (see Methods for detailed classification criteria).

Current PCI strategies for ARCA-LSOV rely heavily on operator experience and ad hoc catheter selection. While conventional guides (Judkins Right, Amplatz Left) are commonly used in standard cases [12], their efficacy decreases for anomalous origin from LSOV and high take-off anomalies [7]. This has led to exploration of alternative approaches, including radial-dedicated catheters (Ikari Left) and left coronary guides repurposed for right coronary use [13].

The Extra Back-Up (EBU) catheter, originally designed for left coronary interventions, offers several theoretical advantages for ARCA-LSOV PCI, including enhanced passive support from contralateral aortic wall contact [14], a wire-modifiable curvature that allows adaptation to various takeoff angles, and a single-catheter strategy that may reduce the need for equipment exchanges.

Despite these potential benefits, systematic evaluation of EBU catheters in ARCA-LSOV PCI remains lacking. This retrospective case series analyzes our center's experience with EBU-guided PCI in 17 consecutive cases of ARCA, focusing on procedure success rates across anatomical subtypes, frequency of catheter exchanges, and safety outcomes including complication rates. Our findings aim to provide practical insights and a structured description of its performance in these challenging anatomies, potentially simplifying the interventional approach while maintaining safety and efficacy.

## Methods

### Study design and setting

This retrospective case series was conducted at a high-volume tertiary referral center. We analyzed our institutional angiography volume from May 2019 to June 2024, which totaled approximately 31,000 procedures. Within this population, a total of 68 patients with anomalous right coronary artery (ARCA) were identified, representing a prevalence of 0.22%. Among this ARCA population, we selected all consecutive patients in whom an EBU catheter was used for either angiography or PCI.

The EBU catheter was employed as a specific technical strategy in 17 patients, representing a substantial subset of the total ARCA cases. The final cohort included 14 patients who underwent PCI, one who had fractional flow reserve (FFR) assessment only, and two who underwent diagnostic angiography without intervention. EBU was selected as the initial guide catheter in 12 cases and as a rescue strategy after unsuccessful attempts with other catheters in the remaining 5 cases.

The study was approved by the Institutional Ethics Committee of the First Affiliated Hospital of Zhengzhou University (IRB No. 2024-KY-0118-272). Due to its retrospective design, the requirement for informed consent was waived. Medical records were accessed for research purposes between 10/07/2024 and 12/08/2024.

### Anatomical classification

ARCA-LSOV cases were categorized into three anatomical subtypes based on angiographic origin relative to the LCA ostium [10]:

Type A: Ostium above the LCA

Type B: Ostium below the LCA

Type C: Midline or commissural origin between the LCA and right sinus

An additional group included patients with high take-off RCA, defined as origin ≥10 mm above the sinotubular junction (Fig 1).

### Procedure protocol

All procedures were performed via transradial access. Operators selected EBU (typically 3.5 or 4.0) based on ostial direction and procedural goals. Adjunctive techniques such as guide extension or microcatheters were employed at operator discretion.

### Data collection and endpoints

Patient demographic data, anatomical classification, and procedural details were collected from institutional records and catheterization reports.

Key outcomes included:

**Primary Endpoint:** Procedural success with EBU catheter, defined as successful RCA engagement, full device delivery, and procedure completion without catheter exchange.

**Secondary endpoints:**

• Need for guide catheter exchange

• Use of guide extension or adjunctive tools

• Catheter-related complications (e.g., ostial dissection, pressure damping)

• Clinical adverse events within 30 days

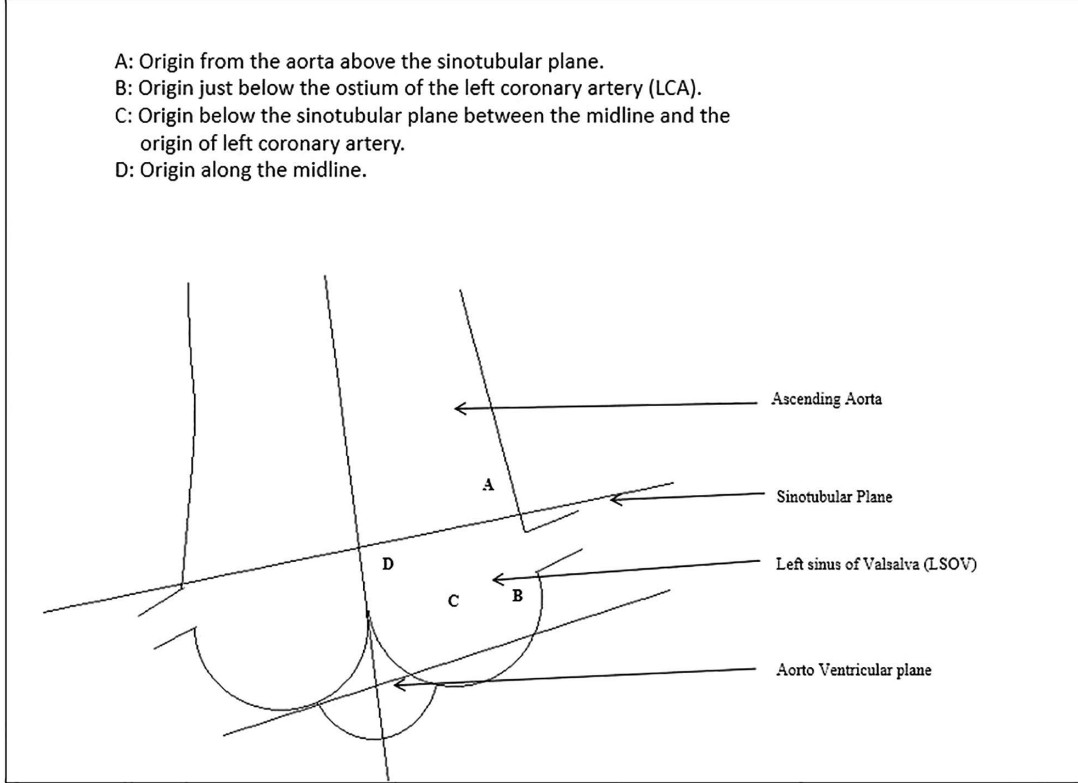

A: Origin from the aorta above the sinotubular plane.
B: Origin just below the ostium of the left coronary artery (LCA).
C: Origin below the sinotubular plane between the midline and the origin of left coronary artery.
D: Origin along the midline.

**Fig 1. A simple anatomical illustration showing anomalous RCA classification according to ostium location (relation to sino-tubular plane and Left main ostium).**

## Results

### Patient and anatomic characteristics

Seventeen patients with angiographically confirmed anomalous right coronary artery (ARCA) underwent coronary angiography or PCI using an EBU catheter. The mean age was 61.3±9.8 years, and 70.6% were male. Hypertension was the most common comorbidity (58.8%), followed by a history of smoking (41.2%), diabetes mellitus (17.6%), and chronic kidney disease (11.8%). Most patients presented with stable angina (58.8%), while 23.5% had non–ST elevation acute coronary syndrome; no cases of STEMI were recorded. Three patients (17.6%) underwent diagnostic angiography or FFR-only assessment without intervention (Table 1).

The majority of ARCA cases (82.4%) originated from the left coronary sinus; classified as Type A, B, or C; while the remaining 17.6% had high take-off anatomy, defined as an origin ≥10 mm above the sinotubular junction (Table 2).

### EBU catheter use and procedural outcomes

The EBU catheter was used as the first-choice guide in 12 patients, with varying success depending on RCA origin type (Table 3). Success was defined as stable engagement, device delivery, and completion of the procedure without catheter exchange.

- **Type A (n = 3):** EBU used first in 2 cases and successfully in both. One additional case used EBU after a failed initial catheter. One patient required guide extension support.

**Table 1. Baseline clinical characteristics of patients undergoing ARCA with EBU catheter.**

| Characteristics | N (%) |
|---|---|
| Total Patients | 17 (100%) |
| Male | 12 (70%) |
| Female | 5 (29.4%) |
| Age, mean ± SD (Years) | 61.3 ± 9.8 |
| Hypertension | 10 (58.8%) |
| Diabetres mellitus | 3 (17.6%) |
| Chronic Kidney Disease | 2 (11.8%) |
| Current or former smoker | 7 (41.2%) |
| Clinical Presentation | N (%) |
| Stable angina | 10 (58.8%) |
| Non- ST elevatin ACS | 4 (23.5%) |
| STEMI | 0 |
| Diagnostic or FFR only | 3 (17.6%) |

**Table 2. Anatomic and procedural features of ARCA cases treated with EBU catheter.**

| Characterisitics | N (%) |
|---|---|
| RCA origin from Left cusp(Type A-C) | 14 (82.4%) |
| Type A (above LCA Ostium) | 3 (17.6%) |
| Type B (below LCA Ostium) | 9 (52.9%) |
| Type C (midline/commissural origin) | 2 (11.8%) |
| High take-off RCA (≥10 mm above STJ) | 3 (17.6%) |
| Procedural Features | |
| CTO | 2 (11.8%) |
| Radial access used | 17 (100%) |
| EBU used as first choice catheter | 12 (70.6%) |
| Use of guide extension catheter | 7 (41.2%) |

**Table 3. EBU catheter performance and utilization by anatomical subtype.**

| Anatomical Subtype | Total Cases | EBU Utilization and Success | Overall EBU Success* | Catheter Conversion from EBU | Guide Extension Used |
|---|---|---|---|---|---|
| Type A | 3 | First: 2/2; Rescue: 1/1 | 3/3 (100%) | 0 | 1 |
| Type B | 9 | First: 5/6; Rescue: 3/3 | 8/9 (88.9%) | 1 (to JL) | 3 |
| Type C | 2 | First: 1/2; Rescue: 0/0 | 1/2 (50%) | 1 (to AL) | 1 |
| High Take-off | 3 | First: 1/2; Rescue: 1/1 | 2/3 (66.7%) | 1 (to SAL) | 2 |
| Total | 17 | First: 9/12; Rescue: 5/5 | 14/17 (82.4%) | 3 | 7 |

*Overall EBU Success: procedural success with EBU catheter (whether as first or rescue), defined as stable engagement, full device delivery, and procedure completion without catheter exchange.

- **Type B (n = 9):** EBU used as first catheter in 6 cases, with 5 successful. One case required conversion to Judkins Left. In three additional B-type cases, EBU was used after failure of other catheters. Three patients required guide extension support. Representative pre- and post-intervention angiographic images of a Type B anomalous RCA PCI using EBU as shown in (Fig 2).

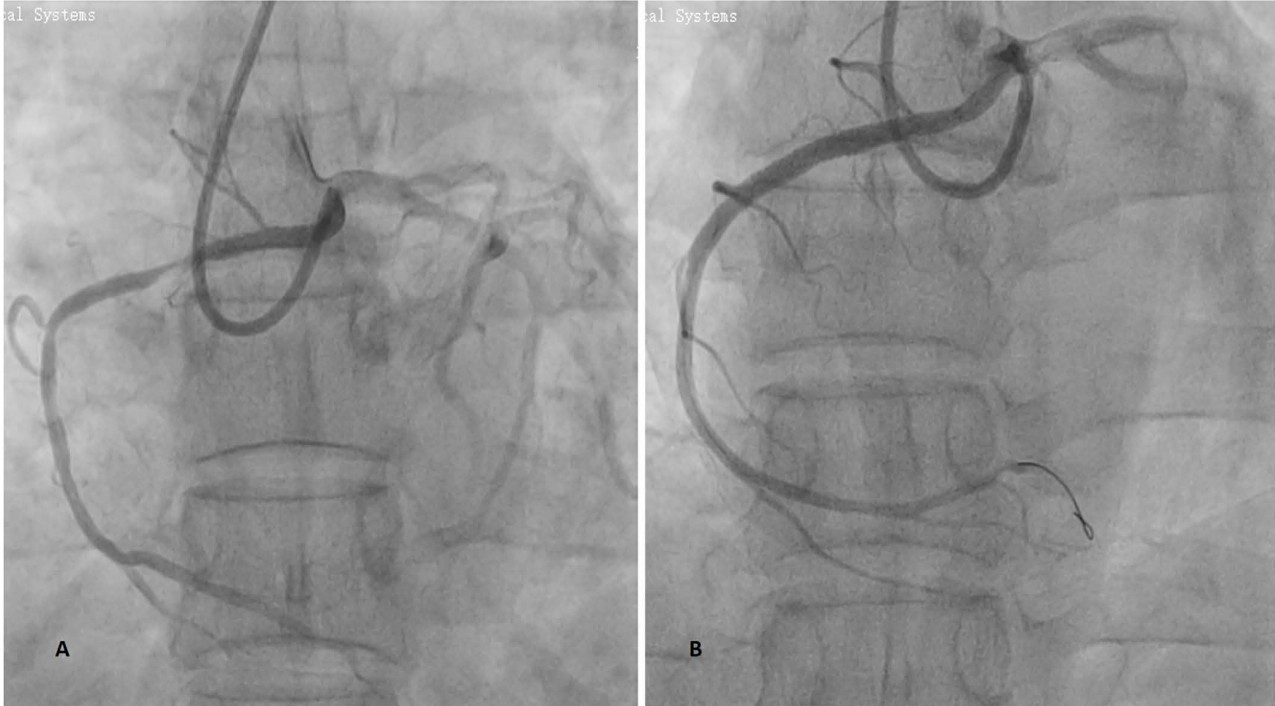

**Fig 2. Procedural angiographic images illustrating EBU catheter use for a Type B anomalous RCA from the left cusp. A** – Angiographic image of Type B anomalous RCA by EBU guide catheter with ostium originating just below the left main ostium showing a proximal to mid segment stenotic lesion; **B** – Post PCI procedure image showing satisfactory outcome after utilizing EBU guide catheter for the procedure.

- **Type C (n = 2):** EBU used first in both cases; one succeeded one required conversion to Amplatz Left. Both cases used guide extension catheters.

- High Take-Off (n = 3): EBU used first in 2 cases, with one success and one requiring conversion. One additional case used EBU after a failed attempt with another catheter. One case used guide extension support.

In summary, the EBU catheter was employed as a second-choice guide in five patients after failed engagement with standard catheters: one with Type A anatomy, three with Type B anatomy, and one with high take-off anatomy.

Based on this usage, the overall procedural success rate with the EBU catheter (whether as a first or subsequent choice) was calculated for each anatomical subtype. Success proportions with 95% confidence intervals were as follows: Type A: 100% (3/3; 95% CI 0.44–1.00), Type B: 88.9% (8/9; 95% CI 0.57–0.98), Type C: 50% (1/2; 95% CI 0.09–0.91), and High-takeoff: 66.7% (2/3; 95% CI 0.21–0.95). Pairwise comparisons using Fisher's exact test found no statistically significant differences in success rates between any single subtype and all others combined (all p-values ≥ 0.228). Furthermore, no significant differences were observed across subtypes for fluoroscopy time, contrast volume, or total procedure time (Kruskal–Wallis p = 0.770, 0.910, and 0.961, respectively).

## Procedural safety

No catheter-induced coronary dissections were observed. One patient experienced a minor in-hospital complication unrelated to catheter selection. One patient required re-intervention PCI for revascularization within thirty postoperative days.

## Discussion

Percutaneous intervention in ARCA presents distinct technical challenges due to ostial eccentricity, unusual take-off angles, and variable alignment with standard catheter geometries. This case series explored the procedural utility of the EBU catheter; originally designed for left coronary interventions, in a consecutive series of ARCA-LSOV cases.

In our technical experience, the EBU catheter was employed across various ARCA anatomical subtypes. We observed that the catheter's design characteristics—particularly its leftward-oriented curve and passive support mechanism—appeared well-suited for engagement in Type A and B anatomies originating near the left coronary cusp. The consistent success we documented in these configurations (100% and 88.9% respectively) suggests this approach merits consideration when encountering such anatomical patterns. Conversely, we found the technique less consistently successful in Type C and high take-off variants, where achieving stable coaxial alignment proved more challenging.

These observations align with prior studies describing the technical complexity of ARCA-LSOV PCI. For instance, Sarkar et al. emphasized shape-specific catheter matching based on ostial origin using FL, FCL, VL, and AL curves tailored to take-off type [10]. Nevertheless, a stepwise and flexible approach to catheter selection remains essential, allowing operators to escalate as needed based on procedural challenges.

The design of the EBU catheter may offer a mechanical advantage in ARCA cases where the origin lies near the left coronary cusp, particularly in Type A and B configurations. Unlike standard right coronary catheters, the EBU's curve is optimized for left coronary engagement, providing a natural alignment with ARCAs that originate close to the LCA ostium. This geometry allows for better coaxial engagement in ostia with an anterior or superior trajectory relative to the aorta. Additionally, the EBU's strong passive support from the contralateral aortic wall can help stabilize the guide, which is critical in tortuous or high-resistance lesions. These structural characteristics explain the high procedural success observed in our Type A and B cases, where the catheter's alignment and backup support synergized with the RCA take-off angle, reducing the need for multiple exchanges or aggressive manipulation.

The EBU catheter's design provides passive support from the contralateral aortic wall, which we found beneficial in certain ARCA anatomies [14]. This contrasts with JR catheters, which may lack sufficient support in tortuous or complex lesions [15], or AL catheters, which offer stronger support but carry an increased risk of ostial injury and deep seating [16]. The Judkins Left (JL) catheter can be repurposed for right coronary anomalies, though its design generally provides less consistent backup support than guides designed for enhanced support [13].

This study aims to identify the scenarios in which the EBU catheter is most effective for ARCA interventions, without promoting a uniform EBU-first strategy. Since performance of other catheters in these cases was not analyzed, the findings are intended to add to the procedural literature and provide real-world insights for interventionists using the EBU catheter in ARCA cases.

In summary, our technical experience describes the use of EBU catheters across varied ARCA anatomical subtypes. We found this approach consistently feasible for Types A and B, while more variable in other configurations. These observations contribute to the collective technical knowledge of guide catheter options for anomalous coronary interventions, emphasizing that multiple approaches may be considered based on individual anatomical characteristics.

## Strength and limitations

Our cohort represents a selective but clinically relevant population; ARCA cases where operators judged the EBU catheter to offer technical advantages. While this excludes cases successfully managed with other catheter types, the five-year collection period and case volume are consistent with the rarity of this anomaly in tertiary centers. Nonetheless, the study is limited by its single-center, retrospective design, selective inclusion of EBU cases, and modest sample size (n = 17), which restrict generalizability, preclude a comparative analysis with other catheters, and limit subgroup analysis. Anatomical classification relied on operator assessment without core lab validation or advanced imaging, and follow-up was confined

to procedural and short-term outcomes. Despite these limitations, the study provides meaningful procedural insights into the use of EBU catheters across varied anatomical subtypes.

## Author contributions

**Conceptualization:** Milan Sigdel, Zhao Ruixue.

**Data curation:** Mikias Legesse Gebremedhin.

**Formal analysis:** Mikias Legesse Gebremedhin.

**Investigation:** Mikias Legesse Gebremedhin.

**Methodology:** Mikias Legesse Gebremedhin, Milan Sigdel.

**Project administration:** Milan Sigdel.

**Resources:** BinBin Du.

**Supervision:** Milan Sigdel, BinBin Du, Yanzhou Zhang.

**Validation:** Mikias Legesse Gebremedhin, Milan Sigdel, Zhao Ruixue, Yanzhou Zhang.

**Visualization:** Yanzhou Zhang.

**Writing – original draft:** Mikias Legesse Gebremedhin.

**Writing – review & editing:** Mikias Legesse Gebremedhin.

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
