## [Decision Letter · Decision Letter 0]

31 Jul 2025

PONE-D-25-28613What Role Should EBU Catheters Play in the Interventional Approach to Anomalous Right Coronary Arteries?PLOS ONE?

Dear Dr. Gebremedhin,

We look forward to receiving your revised manuscript.

Kind regards,

Ali Sheikhy

Guest Editor

PLOS ONE

Journal Requirements:

2. In the online submission form, you indicated that [Data will be available from the corresponding author upon reasonable request.].

4. Please include your tables as part of your main manuscript and remove the individual files.

Additional Editor Comments:

Thank you for submitting your manuscript titled “What Role Should EBU Catheters Play in the Interventional Approach to Anomalous Right Coronary Arteries?”.

The topic addresses a clinically important and technically challenging area in interventional cardiology, PCI in anomalous right coronary artery (ARCA) cases, and focuses on the use of EBU catheters, a tool that is not routinely discussed for this application. The paper presents observational data from a real-world setting, which may be of interest to interventionalists working in complex anatomical scenarios.

However, after a thorough review, I believe the manuscript requires substantial revision before it can be considered further. The concerns are outlined below:

Major Points:

Study Design Limitations

The retrospective, single-center nature of the study and the small sample size (n=17) significantly limit the generalizability of the findings. While the rarity of ARCA is understood, the conclusions currently drawn seem too definitive for what is effectively a descriptive case series.

Lack of a Control or Comparator Group

The manuscript does not include outcomes for alternative catheters such as Judkins or Amplatz, making it difficult to objectively assess the performance of EBU catheters in this setting. Without comparative data, the conclusions about efficacy are observational at best.

Selection Bias and Framing

The study only includes patients in whom EBU was used, which likely introduces selection bias. This should be more explicitly acknowledged, and the framing of EBU as a "preferred strategy" should be softened.

Absence of Statistical Analysis

No formal statistical comparisons or hypothesis testing are presented. While this may reflect the exploratory intent of the study, it reduces the scientific weight of the conclusions. Some minimal statistical analysis (e.g., Fisher’s exact test for success rates across subtypes) could strengthen the manuscript.

Overstatement of Conclusions

The conclusions suggest a clinical algorithm for catheter selection, but this implication is not justified by the data. Please revise to reflect the exploratory nature of your findings and highlight that EBU may be one useful option rather than the preferred approach.

Minor Points:

The manuscript would benefit from language polishing to reduce redundancy and improve clarity.

Figure 2 is of limited utility in its current resolution; consider providing clearer procedural images or removing if not essential.

Please review the formatting for consistency, particularly in tables and figure legends.

A more balanced discussion of prior literature (including limitations of other catheters) would be helpful.

Reviewers' comments:

Reviewer's Responses to Questions

**Comments to the Author**

1. Is the manuscript technically sound, and do the data support the conclusions?

Reviewer #1: Yes

Reviewer #2: Yes

2. Has the statistical analysis been performed appropriately and rigorously?

Reviewer #1: Yes

Reviewer #2: N/A

3. Have the authors made all data underlying the findings in their manuscript fully available?

Reviewer #1: Yes

Reviewer #2: Yes

4. Is the manuscript presented in an intelligible fashion and written in standard English?

Reviewer #1: Yes

Reviewer #2: No

Reviewer #1: This is a well-written retrospective study exploring the utility of Extra Back-Up (EBU) catheters in percutaneous coronary interventions (PCI) for anomalous right coronary artery (ARCA) cases. The topic is relevant and underreported, especially given the technical challenges these anatomical variants present. The authors provide useful subtype-specific procedural data and highlight practical implications for interventional strategy. However, there are several areas where clarity, depth, and methodological rigor could be improved.

1. The small sample size (n=17) limits the generalizability of the findings. While this reflects the rarity of the condition, a more robust discussion of how this limitation affects the study's implications is needed. Expand the limitations section to address potential selection bias, the lack of comparator groups (i.e., performance of other catheters), and how operator experience may influence outcomes.

2. The manuscript does not systematically compare the EBU catheter’s performance to other commonly used guide catheters (e.g., JR, AL, Ikari, etc.). Even if not directly studied, the authors should include more commentary in the Discussion about how EBU compares in the literature or based on their institutional experience.

3. The definitions for success (stable engagement, full device delivery, no catheter exchange) are clear, but it's not always obvious in the tables or text which patients met all components of success. Consider adding a composite procedural success column to Table 3 or clarifying directly in the Results.

4. Also, there are some minor typo errors:

Line 47: “dee to the retrospective nature...” should be “due to the retrospective nature...”

Line 201: Extra period after “achieve..”

Line 130: "10/07/2024 to 12/08/20242024" – check the formatting of the date.

Reviewer #2: Reviewer Comments on the Manuscript: "What Role Should EBU Catheters Play in the Interventional Approach to Anomalous Right Coronary Arteries?"

Congratulations to the authors on this manuscript. The study addresses an important and underrepresented topic in interventional cardiology. It offers valuable technical insights into the use of EBU catheters in managing challenging cases, specifically anomalous right coronary arteries (ARCA). The subject matter is within the scope of the journal and holds clinical relevance.

Please find below my comments and suggestions:

1. Please consider adding the type of study, for example “A Case Series” to the end of the title to clarify the study design and help set expectations for readers.

2. The manuscript requires major English language revision.

3. Ensure all abbreviations are defined upon first use in both the abstract and main text. For example, FFR (Fractional Flow Reserve) should be spelled out the first time it appears.

4. Please clarify the anatomical subtypes of patients in whom the EBU catheter was used as a second-choice guide. This detail will help strengthen the analysis across subgroups.

5. Revise the in-text citation format to match journal style. For example, “[1][2][3]” should be “[1–3]”. Refer to the journal's author guidelines and ensure consistent citation formatting throughout the manuscript.

6. On page 4, line 96, there appears to be a typographical error: “6(Figure 1)”. Please confirm whether this is an erroneous reference or formatting issue.

7. On page 5, line 117, consider revising the word “algorithm” to a more appropriate alternative such as “strategy” or “framework,”.

8. Please define your inclusion and exclusion criteria clearly. For instance, was there an age limit? Also, specify how many total patients underwent coronary angiography during the study period and how many were identified with ARCA.

9. To avoid redundancy, consider explaining the anatomical classification system either in the Methods or the Results section, but not both.

10. Tables were not visible in the version reviewed. Please ensure all tables are included in the final submission and are organized clearly according to anatomical subgroups, with baseline and procedural outcome data shown separately for each group.

11. Please clarify the study type. Since there is no comparison group, this work should be classified as a case series rather than a comparative study.

12. Consider redesigning Figure 1 to enhance visual clarity. A more simplified and illustrative schematic may help readers better understand the anatomical subtypes described.

13. There is a selection bias that should be acknowledged in the limitations section. The study includes only patients in whom the EBU catheter was attempted, and does not provide comparative data on other catheters (e.g., Amplatz, Judkins Left). This limits the generalizability of the findings.

14. Figure 2 was also not viewable. Please ensure that all figures are properly uploaded and embedded in the final version.

**Do you want your identity to be public for this peer review?** For information about this choice, including consent withdrawal, please see our Privacy Policy

Reviewer #1: No

Reviewer #2: No

---

## [Author Response · Author response to Decision Letter 1]

4 Nov 2025

Response to Reviewers

We thank the editors and reviewers for their thoughtful comments and constructive feedback on our manuscript. We have carefully considered all suggestions and have substantially revised the manuscript accordingly. Please find below our point-by-point responses to each comment.Response to Journal Requirements:

Style Requirements: The manuscript has been reformatted to meet PLOS ONE's style requirements.

Data Availability: All underlying data are now available as Supporting Information (S1_Dataset).

Ethics Statement: The ethics statement now appears only in the Methods section.

Tables: All tables are now included within the main manuscript file.

Response to Additional Editor Comments:

Comment: The conclusions were overstated and implied a clinical algorithm.

Response: We have significantly softened the language throughout the manuscript, removing any suggestion of an "algorithm" or "preferred strategy." The conclusions now frame our findings as preliminary insights that support a tailored approach, not a definitive guide. (See Abstract( line 73-75), Introduction(line 110), and Discussion(line 239).

Response to Reviewer #1:

We thank Reviewer #1 for their positive feedback and valuable suggestions.

Comment 1: Expand the limitations section to address selection bias, lack of comparator, and operator experience.

Response: We have expanded the 'Strengths and Limitations' section to explicitly discuss these points, including the selective inclusion of EBU cases and the consequent inability to perform a comparative analysis. (See Limitations section line 256).

Comment 2: Include more commentary on how EBU compares to other catheters in the literature.

Response: We have added a new comparative paragraph in the Discussion that directly compares EBU to Judkins Right, Amplatz Left, and Judkins Left catheters. (See Discussion section line 232-238).

Comment 3: Clarify the composite procedural success in the Results/Table.

Response: We have added a footnote to Table 3 explicitly defining the composite endpoint of procedural success. The results text also now clearly states when success refers to EBU as a first choice versus overall use. (See Table 3 and Results line 190-201).

Comment 4: Correct minor typographical errors.

Response: All noted typos have been corrected throughout the manuscript.

Response to Reviewer #2:

We thank Reviewer #2 for their encouraging words and detailed recommendations.

Comment 1: Add "A Case Series" to the title.

Response: We have amended the title as suggested to: "What Role Should EBU Catheters Play in the Interventional Approach to Anomalous Right Coronary Arteries? A Case Series."

Comment 2: Major English revision.

Response: We have thoroughly polished the manuscript to improve clarity, reduce redundancy, and ensure standard English usage throughout.

Comment 3: Define all abbreviations upon first use.

Response: We have ensured that all abbreviations are defined upon first use.

Comment 4: Clarify anatomical subtypes for second-choice EBU use.

Response: We have clarified this in the Results section, explicitly stating: "one with Type A anatomy, three with Type B anatomy, and one with high take-off anatomy." (See Results line190-192).

Comment 5: Revise in-text citation format.

Response: The citation format has been corrected throughout to use the journal's style (e.g., [1-3]).

Comment 6: Check typo "6(Figure 1)".

Response: The typo has been corrected.

Comment 7: Revise "algorithm" to "strategy" or "framework."

Response: We have replaced "algorithm" with more appropriate terms like "structured description" and "tailored approach" throughout the text.

Comment 8: Define inclusion/exclusion criteria and provide prevalence.

Response: We have clarified the study population in the Methods, stating the total angiogram volume (~31,000), total ARCA patients (68, prevalence 0.22%), and the subset where EBU was used (17). (See Methods line 115-117).

Comment 9: Avoid redundancy in anatomical classification.

Response: We thank the reviewer for this suggestion. We have revised the manuscript to eliminate redundancy. The Introduction now provides a brief conceptual overview of the anatomical classification (line 89-91), while the detailed, operational criteria used for analysis are presented exclusively in the Methods section.

Comment 10: Include tables in the main manuscript.

Response: All tables are now included in the main manuscript file.

Comment 11: Clarify as a case series.

Response: The manuscript is now consistently described as a "case series" in the title and throughout.

Comment 13: Acknowledge selection bias.

Response: We have explicitly acknowledged the selection bias in the Limitations section (line 254).

Additional Statistical Analysis:

As suggested, we have added comprehensive statistical analysis including:

Descriptive statistics with 95% confidence intervals for success rates by anatomical subtype, Fisher's exact tests for categorical comparisons , Kruskal-Wallis tests for continuous variables across subtypes

(See Results section line 191-199)

We believe these revisions have substantially strengthened the manuscript and addressed all concerns raised during the review process. Thank you for considering our revised submission.

---

## [Decision Letter · Decision Letter 1]

14 Dec 2025

What Role Should EBU Catheters Play in the Interventional Approach to Anomalous Right Coronary Arteries? A case series

PONE-D-25-28613R1

Dear Dr. Gebremedhin,

We’re pleased to inform you that your manuscript has been judged scientifically suitable for publication and will be formally accepted for publication once it meets all outstanding technical requirements.

Kind regards,

Dr Redoy Ranjan, MS (CV&TS), Ch.M. (Edin), PhD

Academic Editor

PLOS One

Additional Editor Comments (optional):

Reviewers' comments:

Reviewer's Responses to Questions

**Comments to the Author**

Reviewer #2: All comments have been addressed

Reviewer #3: All comments have been addressed

2. Is the manuscript technically sound, and do the data support the conclusions?

Reviewer #2: Yes

Reviewer #3: Yes

3. Has the statistical analysis been performed appropriately and rigorously?

Reviewer #2: Yes

Reviewer #3: Yes

4. Have the authors made all data underlying the findings in their manuscript fully available?

Reviewer #2: Yes

Reviewer #3: Yes

5. Is the manuscript presented in an intelligible fashion and written in standard English?

Reviewer #2: No

Reviewer #3: Yes

Reviewer #2: The revised title appropriately reflects a case series, avoiding overstated claims. The success proportions (with 95% CI) and statistical comparisons (Fisher, Kruskal-Wallis) now strengthen the analytic rigor. Please be more cautious with the tables, such as the percentages of males and females that don't match. Also there are some spaces in some of the rows. Otherwise, the manuscript is significantly improved and addresses my concerns.

Reviewer #3: This exploratory study holds significant value by contributing a novel technical perspective to addressing the complex challenge of ARCA. The authors have appropriately addressed the previously noted concerns, including enhanced descriptive details and adherence to scholarly standards. Given the comprehensive clarification of the study’s limitations, this paper is now suitable for publication and will provide clinicians in the field with practical preliminary insights and a useful reference.

**Do you want your identity to be public for this peer review?** For information about this choice, including consent withdrawal, please see our Privacy Policy

Reviewer #2: No

Reviewer #3: No

---

## [Editor Report · Acceptance letter]

PONE-D-25-28613R1

PLOS One

Dear Dr. Gebremedhin,

I'm pleased to inform you that your manuscript has been deemed suitable for publication in PLOS One. Congratulations! Your manuscript is now being handed over to our production team.

Kind regards,

on behalf of

Dr. Redoy Ranjan

Academic Editor

PLOS One